# Conditional Structure Generation through Graph Variational Generative Adversarial Nets

**Carl Yang,**\* **Peiye Zhuang, Wenhan Shi, Alan Luu, Pan Li**
University of Illinois at Urbana Champaign, Urbana, IL 61801
{jiyang3, peiye, wenhans2, alanluu2, panli2}@illinois.edu

## Abstract

Graph embedding has been intensively studied recently, due to the advance of various neural network models. Theoretical analyses and empirical studies have pushed forward the translation of discrete graph structures into distributed representation vectors, but seldom considered the reverse direction, *i.e.*, generation of graphs from given related context spaces. Particularly, since graphs often become more meaningful when associated with semantic contexts (*e.g.*, social networks of certain communities, gene networks of certain diseases), the ability to infer graph structures according to given semantic conditions could be of great value. While existing graph generative models only consider graph structures without semantic contexts, we formulate the novel problem of conditional structure generation, and propose a novel unified model of graph variational generative adversarial nets (CONDGEN) to handle the intrinsic challenges of flexible context-structure conditioning and permutation-invariant generation. Extensive experiments on two deliberately created benchmark datasets of real-world context-enriched networks demonstrate the supreme effectiveness and generalizability of CONDGEN.

## 1 Introduction

Graphs (networks) provide a generic way to model real-world relational data, such as entities in knowledge graphs, users in social networks, genes in regulatory networks, *etc*. It is thus critical to study the generation of graph structures, which is fundamental for the understanding of their underlying functional components and creation of meaningful structures with desired properties. Nowadays, contextual data like attributes and labels are becoming ubiquitous in networks [44], the rich semantics of which may well correspond to particular graph structures. This brings up a natural but challenging question: Can we generate graph structures *w.r.t.* given semantic conditions?

In this work, we propose and study the novel problem of conditional structure generation, whose goal is to learn and generate graph structures under various semantic conditions indicated by contextual attributes or labels in the networks. Figure 1 shows a toy example of biomedical networks, where the interactions of certain genes and proteins may follow related but different patterns for individuals with different diseases (*e.g.*, cancers in different body parts and stages). Due to limited observations, network data of some diseases may be more scarce (only one network observed for *Case 1*) or totally missing (no network observed for *Case 2*), while those of other closely related diseases are more available (2-3 networks observed for other cases). Since the diseases are semantically related, their corresponding gene networks may well share certain graph structures. Thus, by efficiently exploring the possible correspondence between network contexts and structures, an ideal model should be able to generate more similar graphs for conditions with scarce observed graphs (*Task 1*), and generate meaningful novel graphs for conditions without any observed graphs (*Task 2*). The problem is important because the generated networks can be valuable in various subsequent studies

---

*\*Corresponding author.*

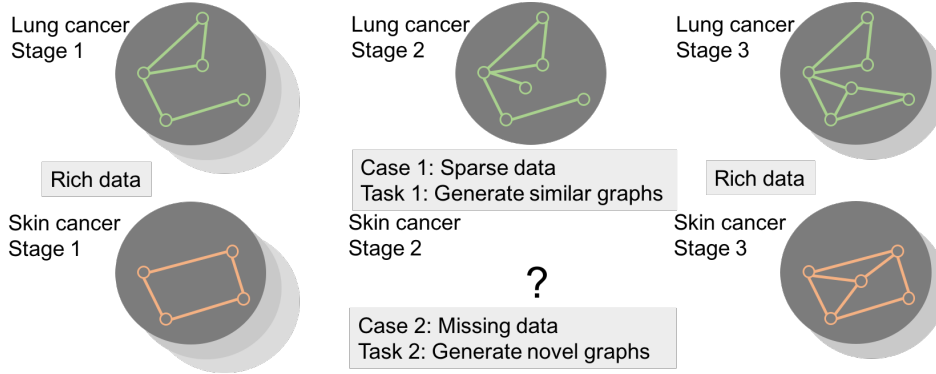

Figure 1: **Toy example of conditional structure generation**: Real-world networks nowadays are often associated with correlated semantic attributes/labels. This allows us to explore the possible correspondence between graph contexts and structures, which can be leveraged to generate structures for graphs with certain semantic contexts that are hardly observed.

such as the understanding and prediction of disease development. It is also general, if we consider vast other examples such as in social networks, where users in different communities may share certain connection patterns, and in knowledge graphs, where different types of entities may be related in particular ways.

Existing works on network generation cannot flexibly handle various semantic conditions. Specifically, earlier probabilistic graph models can only generate networks with limited pre-assumed properties like random links [10], small diameters [41], power-law distribution [2], *etc*. Recent works based on neural networks can generate graphs with much richer properties learned on given graphs. However, they either only work with single graphs and fixed sets of nodes [18, 4, 7, 25, 52, 12], or model single representative sets of graphs which essentially belong to the same semantic group [47, 36, 16, 24, 46]. Only [24] mentions the ability of conditional generation, but the conditions in their setting are direct graph properties such as number of nodes, which is fundamentally different from the semantic conditions as we consider in this work. Moreover, none of the existing methods really solve the fundamental challenge of graph permutation invariance [26, 42, 15, 48, 30] during their translation between graph structures and representations, due to the facts that their embedding spaces or generated graphs are essentially not permutation-invariant (Sec. 2), so they tend to generate different graphs given the same input graphs with permuted node orders (Sec. 4).

Thanks to the surge of deep learning [20, 27], many successful neural network models like skip-gram [28] and CNN [17] have been studied for graph representation learning [31, 11, 38, 19, 39]. Among them, graph convolutional neural networks (GCN) [19] has received extensive theoretical analyses and empirical studies recently [26, 42, 14, 5, 23], due to its proved ability to encode nodes, (hyper)links or whole graphs into a permutation-invariant space. However, how to map the distributed vectors back to graphs in a permutation-invariant manner still remains an open problem. Particularly, the graph variational autoencoder (GVAE), as the direct application of GCN for graph generation [18], still only models single networks with fixed sets of nodes (with fixed orders), thus cannot handle flexible semantic conditions and permutation invariance.

In this work, to address the essential challenges of *flexible context-structure conditioning* and *permutation-invariant graph generation* in conditional structure generation, we propose the novel model of CONDGEN, which is essentially a neural architecture of graph variational generative adversarial nets. It fully leverages the well developed GCN model by further collapsing the node encoding into permutation-invariant graph encoding through variational inference on the conjugate latent distributions, which naturally allows flexible graph context-structure conditioning. To further guarantee permutation-invariant graph decoding/generation, GCN is leveraged again to construct a graph discriminator before the computation of graph reconstruction loss in the standard encoder-decoder framework. This allows the graph generator to explore graphs of variable sizes and arbitrary node orders, which is critical for the capturing of essential graph structures. Finally, for efficient and robust model training, we let the GCNs in graph encoder and discriminator share parameters to enforce mapping consistency between the graph context and structure spaces and avoid the encoder collapse.

To fully demonstrate the value of conditional structure generation and the power of our proposed CONDGEN model, we create two benchmark datasets of real-world context-enriched networks and design a series of experiments to evaluate CONDGEN against several state-of-the-art graph generative models properly adapted to the same setting. Through close comparisons over various graph properties and careful visual inspections, we comprehensively show the supreme effectiveness and generalizability of CONDGEN on conditional structure generation.

## 2 Graph Variational Generative Adversarial Nets

### 2.1 Problem Formulation

We focus on the novel problem of conditional structure generation. We are given a set of graphs $G = \{G_1, G_2, \ldots, G_n\}$, where $G_i = \{V_i, E_i\}$ corresponds to a particular *graph structure* described by the set of nodes $V_i$ and the set of edges $E_i$. Since graphs nowadays are often contextualized with certain semantic attributes or labels of interest, we construct a condition vector $C_i$ for each graph $G_i$, which describes some particular simple *graph contexts* of $G_i$ (examples are shown in the data preparation in Sec. 4). We leave the exploration of more complex contexts as future work.

In this work, we aim to explore and model the possible context-structure correspondence on graphs. That is, by training a model $\mathcal{M}$ on a set of graphs with certain conditions (*i.e.*, $\mathcal{T} = \{G_i, C_i\}_{i=1}^n$), we hope to (1) given a seen condition $C \in \mathcal{T}$, generate more graphs $G$ mimicking the structures of those in the training set $\mathcal{T}$, and (2) given an unseen condition $C \notin \mathcal{T}$, generate reasonable novel graphs $G$ that can support similar tasks in $\mathcal{T}$ while providing insight into the unobservable world.

We summarize the essential challenge of conditional structure generation as two folds in the following.

**Requirement 1.** *Flexible context-structure conditioning.* Both the context space, structure space and mapping between the two spaces can be rather complex. Therefore, a model should be able to effectively explore the two spaces and their correspondence based on all context-structure pairs in $\mathcal{T}$. This means the model needs to jointly capture arbitrary contexts and generate graphs of arbitrary sizes and structures. Moreover, a single model has to be trained on arbitrary numbers of context-structure pairs upon availability.

**Remark 1.** Existing graph generative models only consider graph structures and ignore the rich graph contexts associated for structure generation. Moreover, earlier works only model particular families of structures [22, 33], while more recent works mostly consider single graphs with fixed sizes [18, 4, 7, 25, 52, 12]. GraphRNN [47] is the only one we have seen so far that can be trained with a set of graphs and scale up to graphs with hundreds of nodes, but its GRU design with sequential hidden spaces makes it hard to directly apply effective semantic conditioning (as we will discuss more in Sec. 3 and show in Sec. 4).

**Requirement 2.** *Permutation-invariant graph generation.* The structure of a graph $G$ is most commonly represented by an adjacency matrix $A$, where $A_{ij} = 1$ means $v_i$ and $v_j$ are connected and $A_{ij} = 0$ otherwise. However, the representation is not unique. In fact, since there are $n!$ possible permutations for a graph with $n$ nodes, the number of possible adjacency matrices corresponding to the same underlying graph is also exponential. Therefore, a model should be able to efficiently compare the underlying graphs instead of the representations and equalize different representations of the same underlying graphs, essentially achieving permutation-invariance [26, 42, 15, 48, 30].

**Remark 2.** Existing graph generative models are not permutation-invariant. Particularly, models relying on fixed sets of nodes are not permutation-invariant, because there exists no canonical node ordering and the models have to be re-trained whenever the ordering of nodes is changed [18, 12, 36, 7, 25, 24]. Moreover, models that convert between graphs and other structures like node-edge sequences, trees and random walks are also not permutation-invariant, because there is no guarantee of one-to-one mapping between graphs and the selected structures [47, 16, 4, 46, 52].

### 2.2 Proposed Model

We propose CONDGEN, which coherently joins the power of GCN, VAE and GAN for conditional structure generation, and satisfies both requirements above. Figure 2 illustrates the overall architecture of CONDGEN. In the following, we introduce the motivations and details of our model design.

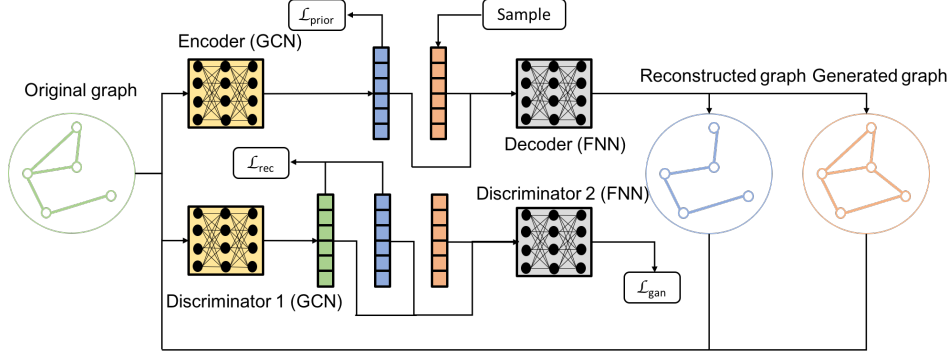

Figure 2: **Overall framework of CONDGEN**: The upper part is a graph variational autoencoder, where we collapse the node embeddings into a single graph embedding, so as to enable flexible graph context-structure conditioning and allow training/generating of graphs with variable sizes. The lower part makes up for a graph generative adversarial nets, where we leverage GCN to guarantee permutation-invariant graph encoding, generation and comparison for reconstruction. Parameters in the decoder and generator networks as well as those in the two GCN networks in the encoder and discriminator are shared to further boost efficient and robust model inference.

Given the two requirements, we get inspiration from recent works on GCN, which is promising in calculating representative and permutation-invariant graph embedding [26, 42]. It is thus natural to think of a permutation-invariant graph encoder-decoder framework by leveraging GCN and enable flexible conditioning through variational inference [37]. In fact, [18] proposed a VAE framework for graph generation soon after the invention of GCN. However, they only consider learning on a single graph $G = \{V, E\}$ and generating/reconstructing links $E$ on the fixed set of nodes $V$, thus failing to meet both requirements for conditional structure generation.

In this work, we apply a small but necessary trick to the original GVAE framework in [18], *i.e.*, latent space conjugation, which effectively converts node-level encoding into permutation-invariant graph-level encoding, and allows learning on arbitrary numbers of graphs and generation of graphs with variable sizes. Particularly, given a graph $G = \{V, E\}$, since we consider available node contents as semantic conditions, we regard $G$ as a plain network with the adjacency matrix $A$ and generate node features $X = X(A)$ as the standard $k$-dim spectral embedding[2] based on $A$. As suggested by reviewers, we later also experiment with replacing spectral embedding by Gaussian random vectors, which leads to significant reduce in runtime and comparable model performance, thanks to the representative and permutation-invariant structure encoding of GCN (details in Sec. 4).

Following [18], we introduce the stochastic latent variable $Z$, which can be inferred from $X$ and $A$ as $q(Z|X, A) = \prod_{i=1}^{n} q(\mathbf{z}_i|X, A)$. $\mathbf{z}_i \in Z$ can be regarded as the *node embedding* of $v_i \in V$. Different from [18], we use a single distribution $\bar{\mathbf{z}}$ to model all $\mathbf{z}_i$'s by enforcing

$$q(\mathbf{z}_i|X, A) \sim \mathcal{N}(\bar{\mathbf{z}}|\bar{\mu}, \text{diag}(\bar{\sigma}^2)), \text{ where } \bar{\mu} = \frac{1}{n}\sum_{i=1}^{n}\mathbf{g}_\mu(X, A)_i, \bar{\sigma}^2 = \frac{1}{n^2}\sum_{i=1}^{n}\mathbf{g}_\sigma(X, A)_i^2, \quad (1)$$

where $\mathbf{g}(X, A) = \tilde{A}\text{ReLU}(\tilde{A}XW_0)W_1$ is a two-layer GCN model. $\mathbf{g}_\mu(X, A)$ and $\mathbf{g}_\sigma(X, A)$ compute the matrices of mean and standard deviation vectors, which share the first-layer parameters $W_0$. $\mathbf{g}(X, A)_i$ is the $i$th row of $\mathbf{g}(X, A)$. $\tilde{A} = D^{-\frac{1}{2}}AD^{-\frac{1}{2}}$ is the symmetrically normalized adjacency matrix of $G$, where $D$ is its degree matrix with $D_{ii} = \sum_{j=1}^{n} A_{ij}$.

The trick of latent space conjugation leads to the modeling of $\bar{\mathbf{z}}$, which essentially is the mean of $\mathbf{z}_i$ over $G$, and thus can be regarded as the *graph embedding* of $G$. While straightforward, the introduction of $\bar{\mathbf{z}}$ is critical for conditional structure generation, because (1) it allows the model to generate graphs of variable sizes and be trained on set of graphs; (2) it enables graph-level variational inference and flexible context-structure conditioning; (3) it guarantees permutation-invariant graph encoding. We discuss about these three advantages in details in the following.

Firstly, by individually modeling the embedding $\mathbf{z}_i$ of each node $v_i \in V$ with separate latent distributions, [18] can only generate links among the fixed set of nodes $V$, whereas we can generate graphs of an arbitrary size $m$ by sampling $\mathbf{z}_i$ for $m$ times from the shared distribution of $\bar{\mathbf{z}}$. Secondly, according to [29], a conditional GVAE can be directly constructed by concatenating ($\odot$) the condition vector $C$ to both $X$ and $\bar{\mathbf{z}}$ during training and to $\mathbf{z}_i$'s sampled from $\bar{\mathbf{z}}$ during generation. Finally, since $\mathbf{g}(X, A)$ is permutation-invariant (*i.e.*, $\forall P \in \{0,1\}^{n \times n}$ as a permutation matrix, $\mathbf{g}(PX, PAP^T) = P\mathbf{g}(X, A)P^T$ [45]), $\bar{\mathbf{z}}$, $\bar{\mu}$ and $\bar{\sigma}$ are also permutation-invariant (*i.e.*, $\sum_{i=1}^{n} \mathbf{g}(PX, PAP^T)_i = \sum_{i=1}^{n} [P\mathbf{g}(X, A)P^T]_i = \sum_{i=1}^{n} \mathbf{g}(X, A)_i$). It thus guarantees that $\bar{\mathbf{z}}$ is indistinguishable if $A$ is permutated.

Besides this difference, after sampling a desirable number of $\mathbf{z}_i$'s, to improve the capability of the graph decoder, we append a few layers of fully connected feedforward neural networks $\mathbf{f}$ to $\mathbf{z}_i$ before computing the logistic sigmoid function for link prediction, *i.e.*,

$$p(A|Z) = \prod_{i=1}^{n} \prod_{j=1}^{n} p(A_{ij}|\mathbf{z}_i, \mathbf{z}_j), \text{ with } p(A_{ij} = 1|\mathbf{z}_i, \mathbf{z}_j) = \sigma(\mathbf{f}(\mathbf{z}_i)^T \mathbf{f}(\mathbf{z}_j)), \quad (2)$$

where $\sigma(z) = 1/(1 + e^{-z})$. We optimize the model by minimizing the minus variational lower bound

$$\mathcal{L}_{vae} = \mathcal{L}_{\text{rec}} + \mathcal{L}_{\text{prior}} = \mathbb{E}_{q(Z|X,A)}[\log p(A|Z)] - D_{\text{KL}}(q(Z|X, A)\|p(Z)), \quad (3)$$

where $\mathcal{L}_{\text{rec}}$ is a link reconstruction loss and $\mathcal{L}_{\text{prior}}$ is a prior loss based on the Kullback-Leibler divergence towards the Gaussian prior $p(Z) = \prod_{i=1}^{n} p(\mathbf{z}_i) = \mathcal{N}(\bar{\mathbf{z}}|\mathbf{0}, \mathbf{I})^n$. The model now consists of a GCN-based graph encoder $\mathcal{E}(A) = \frac{1}{n} \sum_{i=1}^{n} \mathbf{g}(X(A), A)_i$, and an FNN-based graph decoder/generator $\mathcal{G}(Z) = \mathbf{f}(\mathbf{z}_i)^T \mathbf{f}(\mathbf{z}_j)$.

With this modified GVAE, we can compute permutation-invariant graph encoding and generate graphs of variable sizes under different conditions. However, the graph generation process is still not permutation-invariant, because $\mathcal{L}_{rec}$ is computed between the generated adjacency matrix $A' = \mathcal{G}(Z)$ and the original adjacency matrix $A$, which means $A'$ has to follow the same node ordering as $A$. In an ideal case, if $A' = PAP^T$, $\mathcal{L}_{rec}$ should be zero. This is not the case for the current model, which misleads the generator/decoder to waste its capacity in capturing the $n!$ node permutations, instead of the underlying graph structures.

To deal with this deficiency, we again leverage GCN, by devising a permutation-invariant graph discriminator, which learns to enforce the intrinsic structural similarity between $A'$ and $A$ under arbitrary node ordering. Particularly, we construct a discriminator $\mathcal{D}$ of a two-layer GCN followed by a two-layer FNN, and jointly train it together with the encoder $\mathcal{E}$ and decoder/generator $\mathcal{G}$, by optimizing the following GAN loss of a two-player minimax game

$$\mathcal{L}_{gan} = \log(\mathcal{D}(A)) + \log(1 - \mathcal{D}(A')), \text{ with } \mathcal{D}(A) = \mathbf{f}'(\mathbf{g}'(X(A), A)), \quad (4)$$

where $X$, $\mathbf{g}'$ and $\mathbf{f}'$ are spectral embedding, GCN and FNN, respectively, similarly as defined before. After $\mathbf{g}'$, the encodings $\mathbf{g}'(A)$ and $\mathbf{g}'(A')$ are permutation-invariant (*i.e.*, $\forall A' = PAP^T, \mathbf{g}'(A) = \mathbf{g}'(A')$), and the reconstruction loss $\mathcal{L}_{rec}$ can be simply computed as $\mathcal{L}_{rec} = \|\mathbf{g}'(A) - \mathbf{g}'(A')\|_2^2$, which captures the intrinsic structural difference between $A$ and $A'$ regardless of the possibly different node ordering.

At this point, we find our model closely related to the recently popular framework of VAEGAN [21, 13, 34]. Similarly to their observations, we find it beneficial to include two sources of generated matrix $A'$, *i.e.*, one from the sampled graph encoding $Z_s$ *w.r.t.* the prior distribution, and another from the computed graph encoding $Z_c = \mathcal{E}(A)$, and redefine the GAN loss as

$$\mathcal{L}_{gan} = \log(\mathcal{D}(A)) + \log(1 - \mathcal{D}(\mathcal{G}(Z_s))) + \log(1 - \mathcal{D}(\mathcal{G}(Z_c))). \quad (5)$$

Different from VAEGAN, and motivated by the powerful framework of CycleGAN [51], we further aim to apply additional constraints to the framework to enforce mapping consistency between the context and structure spaces. Particularly, we find it beneficial to share parameters in the two GCN modules $\mathbf{g}$ and $\mathbf{g}'$, which essentially requires that the generated graph $A'$ can be brought back to the latent space of graph encoding with contexts $Z \odot C$ by the same encoder $\mathbf{g}$ that maps the original graph $A$ to the space of $Z \odot C$. Besides, in practice, it may also help prevent the encoder from occasional collapse due to the overwhelmingly powerful decoder/generator [1], when $\mathcal{E}$ keeps yielding the same noise $Z$ for different input $A$, but $\mathcal{G}$ manages to overfit the training data by generating the correct $A'$ solely based on the condition vector $C$. In this case, the model degrades into a conditional GAN [29], which is harder to train without $\mathcal{E}$ functioning as expected.

## 2.3 Training Details

We jointly train the encoder $\mathcal{E}$, decoder/generator $\mathcal{G}$ and discriminator $\mathcal{D}$ by optimizing the following combined loss function

$$\mathcal{L}_{\text{CONDGEN}} = \mathcal{L}_{rec} + \lambda_1 \mathcal{L}_{prior} + \lambda_2 \mathcal{L}_{gan}, \tag{6}$$

where $\lambda_1$ and $\lambda_2$ are tunable trade-off hyperparameters. As suggested in [21], it is important not to update all model parameters *w.r.t.* the combined loss function. Particularly, we use the following parameter updating rules for in each training batch

$$\theta_E \xleftarrow{+} -\nabla_{\theta_E}(\mathcal{L}_{rec} + \lambda_1 \mathcal{L}_{prior}), \; \theta_G \xleftarrow{+} -\nabla_{\theta_G}(\mathcal{L}_{rec} - \lambda_2 \mathcal{L}_{gan}), \; \theta_D \xleftarrow{+} -\nabla_{\theta_D}\lambda_2 \mathcal{L}_{gan}. \tag{7}$$

# 3 Connections to Existing Works

## 3.1 Graph Embedding

Graph embedding studies the task of computing distributional representations for graph data, where the major challenge lies in the lack of canonical node ordering and flexible context structures. While traditional embedding methods often resort to computations in the spectral domain [6, 3, 35], recent advances in neural networks and deep learning have shed new light on efficient approximations to the heavy spectral computations [31, 38, 11, 32, 19, 8, 43]. Among the many recently developed neural network based graph embedding algorithms, GCN [19] has been intensively studied and shown promising in calculating representative and permutation-invariant graph embedding [23, 45, 5]. Particularly, we harvest the nice properties of GCN illustrated by the following bound

$$\alpha d(G, G') \le d(\mathcal{E}(A), \mathcal{E}(A')) \le \beta d(G, G'), \tag{8}$$

where $d(G, G')$ is the structural difference between two graphs $G$ and $G'$, such as edit distance, $d(\mathcal{E}(A), \mathcal{E}(A'))$ is the distance between the encodings of $G$ and $G'$, such as $\ell_2$ distance, $\alpha$ and $\beta$ are two constants ($\alpha \le \beta$). The first inequality guarantees representativeness of the encoding, while the second guarantees permutation invariance. Properly trained GCN is expected to approximate a tight bound with $\alpha \simeq \beta$ [26, 42], which is thus ideal for our tasks of representative and permutation-invariant graph encoding and discrimination.

## 3.2 Graph Generation

Although much work has been done regarding permutation-invariant graph embedding, how to generate graphs in a permutation-invariant manner is still an open problem. We formulate the requirements of permutation-invariant graph embedding/encoding and generation/decoding as follows, $\forall P \in \{0, 1\}^{n \times n}$ as a permutation matrix

$$\text{Encoding: } \mathcal{E}(PAP^T) = \mathcal{E}(A); \quad \text{Decoding: } \mathcal{L}(A', A) = \mathcal{L}(PA'P^T, A). \tag{9}$$

Most existing graph generation models like [18, 12, 7, 40, 25, 52] are only generating links among fixed set of nodes, and thus do not satisfy Eq. 9. Similar to our leverage of GCN, GraphVAE [36] comprises of a graph encoder of GCN that satisfies the first part of Eq. 9 and a decoder outputting a symmetric adjacency matrix. Since the nodes of the output graph may have arbitrary ordering, to satisfy the second part of Eq. 9, an expensive graph matching algorithm ($O(n^4)$) must be employed before the graph reconstruction loss can be calculated. To avoid explicit matching between generated and original graphs, NetGAN [4] is proposed to convert graphs into biased random walks and learn the generation of walks instead of graphs. However, although much more efficient than GraphVAE, NetGAN still can only model single graphs with fixed sizes. To learn with multiple graphs and generate graphs with variable sizes, GraphRNN [47] is proposed to model graph generation as a node and edge insertion sequence with RNN. However, since RNN models a series of dynamic hidden spaces, it is hard to directly apply conditioning by concatenation. Moreover, due to the lack of one-to-one mapping between graphs and node-edge sequences, both encoding and decoding processes of GraphRNN do not satisfy Eq. 9.

### 3.3 Deep Generative Models

In order to enable flexible conditioning and permutation-invariant generation, our model borrows ideas from VAEGAN [21, 13, 34], which augments the standard VAE model with a discriminator to enforce style similarity and reduce image blurring. Different from VAEGAN, we incorporate a GCN based discriminator to learn a loss function that is both discriminative and permutation-invariant. Moreover, a key aspect of learning this loss function involves cycle consistency, a concept first introduced by CycleGAN [51] to learn a translation mapping $G$ between images in a source space $X$ to images in a target space $Y$. Since $G$ is under-constrained due to the absence of aligned $X, Y$ examples, it is paired with an inverse mapping $F$ from the target space $Y$ to the source space $X$, where a cycle consistency loss is introduced to enforce similarity between $X$ and $F(G(X))$. Our model leverages the idea of cycle consistency by encoding both the original graph $A$ and generated graph $A'$ into the same context space $Z \odot C$ where the GAN loss can then be calculated. Particularly, we require $\mathcal{E}(\mathcal{G}(\mathcal{E}(A)))$ to be close to $\mathcal{E}(A)$ by sharing the parameters in the two GCN modules (*i.e.*, $\mathbf{g}$ in the encoder $\mathcal{E}$ and $\mathbf{g}'$ in the discriminator $\mathcal{D}$) and computing the reconstruction loss $\mathcal{L}_{rec}$ and GAN loss $\mathcal{L}_{gan}$ after $\mathbf{g}'$, thus enforcing cycle consistency between the context and structure spaces. Finally, the idea of enabling permutation-invariance in an encoder-decoder system also applies to problems like set generation, where we notice the recent development of models that achieve it by backpropagating through an order-invariant set encoder [49], which can be further improved with more sophisticated set pooling methods [50].

## 4 Experimental Evaluations

We create two real-world context-rich network datasets and conduct thorough experiments to demonstrate the effectiveness and generalizability of CONDGEN in conditional structure learning. All code and data used in our experiments have been made available on GitHub[3].

**Datasets.** Since we are the first to consider the novel but important problem of conditional structure generation, there is no existing dataset for evaluation. To this end, we created two benchmark datasets, *i.e.*, a set of author citation networks from DBLP[4] and a set of gene interaction networks from TCGA[5].

From DBLP, we create a set of 72 ($8 \times 3 \times 3$) author networks, each associated with a 10-dim condition vector. The nodes are the first authors of research papers published in 8 conferences, *i.e.*, NIPS and ICML (representing the ML community), KDD and ICDM (DM), SIGIR and CIKM (IR), SIGMOD and VLDB (DB). Then each of the 8 groups of authors are further divided into 3 subgroups by the number of total publications (1-10, 10-30, 30+), representing the productivity of authors. Finally three networks are created for each of the 24 sets of authors, by adding in the citation links created in different time period (1990-1999, 2000-2009, 2010-2019). Thus, the 10-dim condition vector is a concatenation of a 8-dim one-hot vector denoting the conferences, and a 2-dim integral vector denoting the level of productivity and link creation time (each with three values 0, 1, 2). The average numbers of nodes and edges in the author networks are 109 and 186, respectively.

From TCGA, we create a set of 54 ($6 \times 3 \times 3$) gene networks, each associated with a 8-dim condition vector consisting of a 6-dim one-hot encoding of cancer primary sites (brain, liver, lung, ovary, skin, and kidney) and a 2-dim integral vector denoting age of diagnosis (30-57, 58-69, 70-90) and stage approximated by days-to-days (0-400, 400-800, 800-8000). For each faceted search with a particular combination of primary site, age-at-diagnosis, and days-to-death filters, a gene correlation network was created using a gene expression matrix constructed from the first 10 RNA-Seq FPKM files. From each RNA-Seq FPKM matrix $M$, a transformed matrix $N = log_{10}(M + 0.5 \times min(M))$ was created and then filtered for genes with a unique entrez ID and vector representation [9]. Finally, a gene correlation network was constructed using pearson correlation with $p$-value threshold 0.01. The average numbers of nodes and edges in the gene networks are 177 and 1096, respectively.

**Baselines.** Since no baseline is available for the novel task of conditional structure learning, we carefully adapt three state-of-the-art graph generation methods, *i.e.*, GVAE [18], NetGAN [4] and

GraphRNN [47], by concatenating the condition vectors to both the node features of the input graph and the output of the last encoding layer following the standard practice in [29]. To allow a single GVAE or NetGAN model to be trained on a set of graphs, we fix the size of input and output graphs as the largest size of all networks following [36]. As suggested by reviewers, we also construct a variant of CONDGEN by replacing the spectral embedding with Gaussian random vectors of the same sizes to use as input node features to GCN, denoted as CONDGEN(R) (*i.e.*, random vectors) as opposed to CONDGEN(S) (*i.e.*, spectral embeddings).

**Protocols.** To demonstrate the effectiveness and generalizability of CONDGEN, we evaluate both tasks of mimicking similar seen graphs and creating novel unseen graphs. We firstly partition all networks at random by a ratio of 1:1 into training and testing sets. Note that, the testing set includes graphs with both seen and unseen conditions in the training set, so a good model that performs well on the testing set has to effectively capture the context-structure correspondence among graphs with the seen conditions and generalize to graphs with unseen conditions.

| Graphs | Models | LCC | TC | CPL | MD | GINI |
|---|---|---|---|---|---|---|
| **DBLP Seen** | Real | 96.00 | 48.54 | 3.696 | 11.62 | 0.3293 |
| | GVAE | 20.91** | 21.76** | 1.390* | 2.32** | 0.1964** |
| | NetGAN | 21.15** | 22.46** | 1.641** | 2.77** | **0.0568**** |
| | GraphRNN | 6.88* | 69.32** | 1.628** | 7.06** | 0.2446** |
| | CONDGEN(R) | 6.70* | **7.70*** | 1.201* | **1.33** | 0.1232* |
| | CONDGEN(S) | **6.00** | 11.32 | **0.963** | 1.48 | 0.0959 |
| **DBLP Unseen** | Real | 102.50 | 58.21 | 4.982 | 14.29 | 0.3223 |
| | GVAE | 17.40** | 17.02** | 1.521** | 3.53* | 0.2479** |
| | NetGAN | 29.57** | 39.85** | 1.494** | 3.71** | **0.0812** |
| | GraphRNN | 6.43 | 73.21** | 1.305* | 6.43** | 0.1447** |
| | CONDGEN(R) | 9.25* | 10.50 | 1.445** | **1.92** | 0.1418** |
| | CONDGEN(S) | **6.33** | **10.17** | **1.162** | **1.92** | 0.0861 |
| **TCGA Seen** | Real | 177.34 | 8913.20 | 4.171 | 38.27 | 0.4192 |
| | GVAE | 54.82** | 2396.94* | 1.538 | 14.10** | 0.2035** |
| | NetGAN | 32.02** | 3614.61** | 1.702** | 17.61** | 0.1289* |
| | GraphRNN | **16.20*** | 2881.68** | 1.899** | 18.78** | 0.2726** |
| | CONDGEN(R) | 34.42** | 2594.16** | 1.542 | 9.50 | 0.1509** |
| | CONDGEN(S) | 23.72 | **2076.05** | **1.524** | **8.32** | **0.1093** |
| **TCGA Unseen** | Real | 177.91 | 8053.18 | 4.143 | 34.34 | 0.4154 |
| | GVAE | 37.18** | 2768.55** | **1.324*** | 13.03** | 0.1497** |
| | NetGAN | 31.36** | 3557.91** | 1.645* | 18.45** | 0.1277** |
| | GraphRNN | **15.73**** | 2605.73** | 1.859** | 13.55** | 0.2647** |
| | CONDGEN(R) | 27.77* | 3083.81** | 1.362* | 10.86* | 0.1413** |
| | CONDGEN(S) | 23.97 | **2058.95** | 1.522 | **8.68** | **0.1003** |

Table 1: Performance evaluation over compared algorithms regarding several important graph statistical properties. The Real rows include the values of real graphs, while the rest are the *absolute values of differences* between graphs generated by each algorithm and the real graphs. Therefore, smaller values indicate higher similarities to the real graphs, thus better overall performance. We conduct paired $t$-test between each baseline and CONDGEN(S), scores with * and ** passed the significance tests with $p = 0.05$ and $p = 0.01$, respectively.

| Graphs | GVAE | NetGAN | GraphRNN | CONDGEN(R) | CONDGEN(S) |
|---|---|---|---|---|---|
| **DBLP** | 12.8 | 398.6 | 299.5 | 31.5 | 72.3 |
| **TCGA** | 10.9 | 414.0 | 192.4 | 27.6 | 52.1 |

Table 2: Runtimes of training all compared algorithms on the two sets of networks (minutes).

**Performances.** Following existing works on generative models [4, 47, 36], we evaluate the generated graphs through visual inspection and graph property comparison[6]. Our model can flexibly generate

graphs with arbitrary numbers of nodes and edges. For fair and clear comparison, when generating each graph, we set the maximum number of nodes and edges to the same as the real graph for all compared algorithms. As shown in Table 1, the suite of statistics we use measure graphs from different perspectives, and different algorithms often excel at particular ones. Our proposed CONDGEN models constantly rank top with very few exceptions on all measures over both datasets. The advantage of CONDGENon generating graphs with seen conditions in the training set demonstrates its utility in generating more similar graphs for conditions where observations might be sparse, while the edge on unseen conditions indicates its generalizability to semantically relevant conditions where observations are completely missing. The CONDGEN(R) model variant has quite competitive performance with CONDGEN(S), which can be explained by the representative and permutation-invariant structure encoding power of GCN. Due to space limit, we put detailed parameter settings, qualitative visual inspections and in-depth model analyses into the appendix in the supplemental materials.

**Runtimes.** Similar to most neural network models, it is meaningless to compute the exact complexity of CONDGEN, because the actual runtimes mostly depend on the number of training iterations until convergence. To this end, we record the average runtimes for the training of all compared algorithms until convergence on the two sets of networks and present in Table 2. As we can clearly observe, state-of-the-art graph generation algorithms like GraphRNN and NetGAN are rather slow, due to the heavy model of RNN and large number of sampled walks, respectively, while CONDGEN and its base model GVAE are much faster. Since CONDGEN and GVAE are basically a simple GCN model encapsulated in a VAEGAN and VAE framework, respectively, we also find that the memory consumptions of CONDGEN and GVAE are orders of magnitudes lower than GraphRNN and NetGAN. Among the two CONDGEN variants, CONDGEN(S) takes about double runtime as CONDGEN(R), due to the computation of spectral embeddings. While the overhead is not significant, it can get more concerning as the networks become larger, due to the essential $O(n^3)$ complexity of spectral embedding. However, since CONDGEN(R) has quite competitive performance with CONDGEN(S), one can use it as a substitute of CONDGEN(S) when efficiency is more of a concern.

## 5 Conclusion

To the best of our knowledge, this is the first research effort towards the novel but important problem of conditional structure generation. To address the two unique challenges of flexible context-structure conditioning and permutation-invariant structure generation, we design CONDGEN by coherently joining the power of GCN, VAE and GAN networks. We created two real-world datasets including a set of author networks and a set of gene networks associated with semantic conditions, and thoroughly demonstrated the effectiveness and generalizability of CONDGEN in comparison with several state-of-the-art graph generative models adapted to the conditional structure generation setting. We hope our results will inspire following-up research on conditional generative models for graph data, as well as future works on its application to various domains where the generation of semantically meaningful networks can be leveraged to support downstream data analysis and knowledge discovery.

## Acknowledgements

Research was sponsored in part by U.S. Army Research Lab. under Cooperative Agreement No. W911NF-09-2-0053 (NSCTA), DARPA under Agreements No. W911NF-17-C-0099 and FA8750-19-2-1004, National Science Foundation IIS 16-18481, IIS 17-04532, and IIS-17-41317, DTRA HD-TRA11810026, and grant 1U54GM114838 awarded by NIGMS through funds provided by the trans-NIH Big Data to Knowledge (BD2K) initiative (www.bd2k.nih.gov). The results shown in this work are or part based upon data generated by the TCGA Research Network: https://www.cancer.gov/tcga.

## Footnotes

[2]https://scikit-learn.org/stable/modules/generated/sklearn.manifold.SpectralEmbedding.html

[3]https://github.com/KelestZ/CondGen

[4]DBLP source: https://dblp.uni-trier.de/

[5]TCGA source: https://www.cancer.gov/tcga

[6]Statistics we use include LCC (size of largest connected component), TC (triangle count), CPL (characteristic path length), MD (maximum node degree) and GINI (gini index), measuring different properties of graphs.

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
