[Supplementary Material]

# GVGAN: Supplementary Materials

## APPENDIX A: Detailed Parameter Settings

As mentioned, our GVGAN model consists of an encoder, a decoder, and a discriminator.

In the encoder, we use a spectral embedding layer to extract the node features solely based on graph structures. The output of the spectral embedding layer is a $n \times d$, where $d$ is set to 5 on DBLP and 10 on TCGA. We select $d$ as such small values because there are some small graphs especially in the DBLP dataset and the Laplacian eigenvectors corresponding to the first few smallest eigenvalues usually capture the most important graph properties such as number of disconnected components, clustering structures, *etc*. A graph convolution layer follows afterwards with the output size of 16. We notice that simply using graph convolution layers tends to give unstable outputs, so we add two linear layers with a one-dimensional batch normalization layer and a ReLU activation layer before obtaining the mean and variance variables. Both mean and variance vectors have a dimension of 6.

In the decoder, we use a graph convolution layer followed by linear layers. We follow the same design of GVAE to reconstruct graphs, *i.e.*, using the encoded vectors generated from the linear layers multiplied by their transpose vectors. Interestingly, we notice that if the dimension of the encoded vectors is large, the output graphs tend be very dense, while a small dimension may lead the graphs having many disconnected components. Thus the selection of 6 is done through vast cross-validation. However, since the set of candidate values is relatively small (we conduct cross-validation on values of 2-10), the hyperparameter selection process is easy to complete.

The discriminator has similar settings as the encoder, *i.e.*, they share the exact same GCN module followed by FNNs with the same design, except that the output here is a single value, differentiating generated graphs from real graphs.

We use Adam optimizers for the training of all modules in the GVGAN with a learning rate of 0.001.

## APPENDIX B: Qualitative Visual Inspections

To interpret the results and different performances of compared algorithms, we conduct careful visual inspections on between the real graphs and generated graphs from different algorithms. We mainly focus on the analysis of DBLP networks, since they are generally smaller, sparser and semantically meaningful (*e.g.*, networks constructed over popular venues like ML conferences, highly productive authors and more recent years tend to be larger and denser, *etc*). To provide a clear view, within the DBLP networks, we further selected graphs with smaller sizes, sparser links, fewer connected components and less triangles, so that visualization with NetworkX[1] does not tend to yield cluttered layouts. Besides graph structures, we also attempted to select graphs with diverse conditions to give a comprehensive analysis on the ability of compared algorithms in capturing graphs with different semantic properties and the correspondence between semantics and structures.

Particularly, we pick out 10 real graphs from the DBLP dataset, 5 with seen conditions during training, and 5 with conditions not seen during training. For generated graphs, since all compared algorithms are not deterministic and tend to generate similar but slightly different graphs given the same condition, we draw three generated graphs by each compared algorithm given each condition.

As we can observe in Figure 1-10 and contemplate:

1. In general, the graphs generated by GVGAN are the most similar to the real graphs, which concretely corroborates our model design.

2. Our adaptions of baselines into the conditional structure generation scenario are effective, because all baselines also managed to capture the various graph structures and the semantic-structure correspondence to some extent and are able to generate different graphs based on given conditions, while each baseline algorithm fails in certain cases.

3. GVAE tends to generate graphs with highly skewed degree distributions. We conjecture this is mainly due to its simple mechanism of generating links based on cosine similarity between pairs of nodes which lacks representation capacity, eventually leading to decoder underfitting. Another possible reason lies in its lack of permutation-invariant loss function, which further wastes the decoder capacity in fitting the particular ordering of adjacency matrices rather than the underlying graph structures.

4. NetGAN mostly fails when the graphs become more complex, probably due to the deficiency of random walks in precisely capturing complex graph structures with large sizes. On the contrary, GraphRNN mostly fails when the graphs are simple, where it tends to generate graphs with small scattered components, probably due to its less justified mechanism of terminating the growth of single graph components by predicting EOF with RNN.

## APPENDIX C: In-depth Model Analyses

To understand how our proposed GVGAN model learns to capture the key properties of graphs, we closely evaluate it along training. Since the results are averaged among all networks in the dataset, which exhibits various graph structures, the variances are pretty large and often do not cancel with each other. Interestingly, we find that most graph properties tend to have larger values on real graphs than random graphs, and thus an untrained model often gives lower values on them compared with a well trained model. Nonetheless, GVGAN manages to approach the values of real graphs rapidly after around one hundred of epochs on most graphs.

Figure 11 shows the in-depth model analyses results on the DBLP dataset, while the results on the TCGA dataset follow the similar trends and are thus omitted. Interested readers are encouraged to run our models which are submitted together in the supplementary materials and see how different models behave during training on the novel task of conditional structure generation. Meanwhile, in order to better demonstrate how the generated graphs can be useful in downstream applications, we are conducting more experiments with advanced graph classification and regression tasks, hoping to see that the graphs generated by GVGAN can successfully 'fool' the classification and regression models, providing unlimited structural data under particular conditions of interest that are close to hardly observed or unobservable real graphs.

Condition vector: [0, 0, 0, 0, 0, 1, 0, 0, 2, 3]
Semantics: CIKM, high productivity, 2000-2009
LCC: 76, TC: 5, CPL: 6.415, MD: 10, GINI: 0.3329
Seen during training

(a) Real graph

(b) Generated graphs by GVAE

(c) Generated graphs by NetGAN

(d) Generated graphs by GraphRNN

(e) Generated graphs by GVGAN

Figure 1: **Visual inspection on DBLP author network 1.**

**Condition vector:  [0, 0, 0, 0, 0, 1, 0, 0, 3, 1]**
**Semantics: CIKM, low productivity, 2010-2019**
**LCC: 12, TC: 0, CPL: 2.424, MD: 7, GINI: 0.1806**
**Seen during training**

(a) Real graph

(b) Generated graphs by GVAE

(c) Generated graphs by NetGAN

(d) Generated graphs by GraphRNN

(e) Generated graphs by GVGAN

Figure 2: **Visual inspection on DBLP author network 2.**

**Condition vector:  [0, 1, 0, 0, 0, 0, 0, 0, 2, 1]**
**Semantics: ICDM, low productivity, 2000-2009**
**LCC: 6, TC: 0, CPL: 1.867, MD: 4, GINI: 0.1333**
**Seen during training**

(a) Real graph

(b) Generated graphs by GVAE

(c) Generated graphs by NetGAN

(d) Generated graphs by GraphRNN

(e) Generated graphs by GVGAN

Figure 3: **Visual inspection on DBLP author network 3.**

Condition vector: [0, 1, 0, 0, 0, 0, 0, 0, 2, 2]
Semantics: ICDM, mid productivity, 2000-2009
LCC: 20, TC: 3, CPL: 2.9, MD: 10, GINI: 0.2955
Seen during training

(a) Real graph

(b) Generated graphs by GVAE

(c) Generated graphs by NetGAN

(d) Generated graphs by GraphRNN

(e) Generated graphs by GVGAN

Figure 4: **Visual inspection on DBLP author network 4.**

**Condition vector:  [0, 1, 0, 0, 0, 0, 0, 0, 2, 3]**
**Semantics: ICDM, high productivity, 2000-2009**
**LCC: 122, TC: 14, CPL: 6.014, MD: 12, GINI: 0.3689**
**Seen during training**

(a) Real graph

(b) Generated graphs by GVAE

(c) Generated graphs by NetGAN

(d) Generated graphs by GraphRNN

(e) Generated graphs by GVGAN

Figure 5: **Visual inspection on DBLP author network 5.**

**Condition vector: [0, 0, 0, 0, 1, 0, 0, 0, 1, 2]**
**Semantics: SIGIR, mid productivity, 1990-1999**
**LCC: 30, TC: 2, CPL: 4.191, MD: 5, GINI: 0.2196**
**Unseen during training**

(a) Real graph

(b) Generated graphs by GVAE

(c) Generated graphs by NetGAN

(d) Generated graphs by GraphRNN

(e) Generated graphs by GVGAN

Figure 6: **Visual inspection on DBLP author network 6.**

**Condition vector: [0, 0, 0, 0, 1, 0, 0, 0, 3, 1]**
**Semantics: SIGIR, low productivity, 2010-2019**
**LCC: 12, TC: 1, CPL: 2.379, MD: 6, GINI: 0.1488**
**Unseen during training**

(a) Real graph

(b) Generated graphs by GVAE

(c) Generated graphs by NetGAN

(d) Generated graphs by GraphRNN

(e) Generated graphs by GVGAN

Figure 7: **Visual inspection on DBLP author network 7.**

**Condition vector: [0, 0, 0, 0, 0, 0, 1, 0, 3, 2]**
**Semantics: SIGMOD, mid productivity, 2010-2019**
**LCC: 15, TC: 1, CPL: 3.448, MD: 5, GINI: 0.1556**
**Unseen during training**

(a) Real graph

(b) Generated graphs by GVAE

(c) Generated graphs by NetGAN

(d) Generated graphs by GraphRNN

(e) Generated graphs by GVGAN

Figure 8: **Visual inspection on DBLP author network 8.**

**Condition vector: [0, 0, 0, 0, 0, 0, 0, 1, 3, 2]**
**Semantics: VLDB, high productivity, 2000-2009**
**LCC: 111, TC: 12, CPL: 7.989, MD: 10, GINI: 0.3336**
**Unseen during training**

(a) Real graph

(b) Generated graphs by GVAE

(c) Generated graphs by NetGAN

(d) Generated graphs by GraphRNN

(e) Generated graphs by GVGAN

Figure 9: **Visual inspection on DBLP author network 9.**

**Condition vector: [0, 0, 0, 1, 0, 0, 1, 0, 2, 2]**
**Semantics: NIPS, mid productivity, 2000-2009**
**LCC: 178, TC: 24, CPL: 6.519, MD: 11, GINI: 0.3493**
**Unseen during training**

(a) Real graph

(b) Generated graphs by GVAE

(c) Generated graphs by NetGAN

(d) Generated graphs by GraphRNN

(e) Generated graphs by GVGAN

Figure 10: **Visual inspection on DBLP author network 10.**

Figure 11: **Different graph statistics evaluated along the training of GVGAN on DBLP (averaged between seen and unseen conditions). GVGAN efficiently learns to capture the key properties of graphs and converges to the values of real graphs with only around 100 epochs of training.**

## Footnotes

[1] https://networkx.github.io/