[Reviews · NeurIPS 2019]

Reviewer 1



Originality: The task of the conditional generation of graphs is new, as well as the constraint of permutation invariance, and the flexibility in terms of the generated graph structures (non-fixed set of nodes). The work is a combination of known techniques: a VAE-GAN architecture adapted to graphs, using graph convolutional neural networks and incorporating the permutation invariance constraint. To the best of my knowledge, the literature review is clear and related work adequately cited. Quality: This paper is technically sound and the VAE-GAN-GCN methodology is rigorously described. The authors also provide the code associated with this paper. It is definitely a complete piece of work. Clarity: The paper is clearly written and organized. It provides enough information for even a non-expert reader. Figure 2 is great. I believe that an expert reader can easily reproduce the results. Significance: The contributions are significant for deep generative modeling of graphs.

Reviewer 2



This is an interesting and well-written paper that aims at unifying graph NNs, VAE and GAN. The node ordering study is important. I have a few questions/comments: a. It is not clear to me how the model can generate output graphs with a number of nodes different from the input graph? b. Spectral embedding requires to solve a linear system, which costs n^3 and thus cannot be scaled to large graphs. How do you scale to large graphs? What is the complexity of the learning system? c. Spectral embedding is an important part of the proposed. It should be presented (what is g(A)?) and also explained why it provides independence wrt node re-ordering. d. An important application of graph generation is for molecule generation, which presents a challenging real-world challenge. It would have been good to evaluate the performance of the proposed system wrt [46] which proposed a competitive GNN+GAN+RL molecule generation model.

Reviewer 3



Originality: The work studies the problem of generating graphs given semantic conditions and proposes a new algorithm to conquer technical difficulties by combining or improving established methods. Overall, I think the work is novel and interesting. Quality: To the best of my knowledge, the work is technically sound. All claims are well supported by experimental results. Qlarity: The paper is well organized and clearly written. Significance: The methods of improving GVAE to generate graphs of different sizes and using adversarial learning to ensure permutation invariance are important contributions and may have great impacts on future works in graph generation.

[Author Response · NeurIPS 2019]

We thank the reviewers for their time, insightful judgements, and suggestions for improvements. Our key contributions
include proposing new research directions on conditional graph generation, developing novel generative architectures
for graphs that keep node-order permutation invariance, and creating two benchmark datasets with extensive model
evaluations. We appreciate that the reviewers clearly recognized all these contributions and reached an agreement
on a very positive evaluation. Next, we will address the insightful questions raised by reviewers, related to the
claimed property to generate graphs with different sizes (**Rev2**), initialization in the training procedure (**Rev2**),
additional experiments on molecule generation (**Rev2**) and ablation study (**Rev3**). As **Rev1** accurately summarized our
contributions and proposed only a minor issue that is fairly easy to solve, we next only focus on questions raised by the
other reviewers but will definitely improve the readability of the whole draft to avoid other similar issues.

**Generating graphs with flexible sizes (Rev2).** We have explained how to generate graphs of flexible sizes in the
second last paragraph on page 4 (*Firstly, . . .*). Sorry for the confusion caused and we will make the statements clearer
in the final version. Particularly, since we apply the novel latent space conjugation trick (details on page 4) and compute
a shared distribution $\bar{z}$ for all nodes, we can then draw arbitrary numbers of random samples from $\bar{z}$ to generate graphs
of any sizes.

**Initialization via spectral embedding (Rev2).** Although neural networks training could be sensitive to initialization,
the initialization via spectral embedding is not that crucial in our case. Simply randomized initialization can also
provide fairly good results, although initialization via spectral embedding may slightly help with the convergence
in model training and the robustness of the results. Moreover, as our current experiments depend on medium-size
networks, the runtime induced by computing spectral embedding is only a very small portion of the whole complexity
($\lesssim 10\%$). Given above observations and considering the paper has already made a very broad range of contributions
(from problem settings, to methodology and experiments), we tend to restrict our discussion on spectral-embedding
initialization so that readers may focus on more important points that we aim to emphasize. However, we appreciate
that Rev2 raised this confusion and we will definitely give a more elaborated discussion in our final version.

That being said, we go back to argue for spectral embedding, regarding its complexity and the permutation in-
variance. Actually, the computation of spectral embedding could be much less than $O(n^3)$. Suppose the network
contains $|E|$ edges and the embedding dimension is $k$. Then, spectral embedding via the top-$k$ SVD computation
(Augmented Lanczos Bidiagonalization Algorithm [Baglama et al. 2005]), is with complexity $O(T(|E|k + k^3))$,
where $T$ is the number of iterations that depends on how precise the solution is (typically viewed as a constant in
practice). As $|E| \ll O(n^2)$ and $k \ll n$ for large real networks in practice, the above complexity is much less than
$O(n^3)$. Moreover, as we are considering GCN-based training, each step of backpropagation could be with complex-
ity $O(|E|k)$ (consider GCN has the operation "adjacency matrix $\times$ node embeddings"). So spectral embedding is
at most with the same complexity as GCN training, although the former sounds to be relatively complex because
GCN training is typically accomplished by using much more parallel computation resources than spectral embedding.
Regarding permutation invariance, it is an important and always needed property in graph encoding and decoding
procedure but is not relevant to initialization of node representation vectors. In other words, any types of initial-
ization do not affect permutation invariance (How powerful are graph neural networks [Xu et al. 2019]). To make
it clearer, consider one output of the graph encoding procedure, $\bar{\mu}$ in Eq. 1, for example: For any permutation ma-
trix $P$, $\sum_{i=1}^{n} g_\mu(PX, PAP^T)_i = \sum_{i=1}^{n}(P\tilde{A}P^T\text{ReLU}(P\tilde{A}P^TPXW_0)W_1)_i = \sum_{i=1}^{n}(P\tilde{A}\text{ReLU}(\tilde{A}XW_0)W_1)_i =$
$\sum_{i=1}^{n} g_\mu(X, A)_i$, where we use $P^TP = I$ and $\text{ReLU}(P\cdot) = P\text{ReLU}(\cdot)$. Such equalities are always satisfied for any
$X$ (any node representation vectors).

**Application in molecule generation (Rev2).** We do not aim at molecule generation in this work, but rather general
network generation without the need of domain knowledge. Such flexibility allows our framework to generate networks
for a wide range of domains (e.g., social networks, gene networks in our experiments and probably many others). To the
best of our knowledge, previous works on molecule generation require chemical valency to guide the training procedure,
so evaluating our method and comparing it with other chemical-valency-dependent approaches are unfair. However, it
is an interesting future direction to investigate how to incorporate domain constraints like chemical valency into our
framework.

**Ablation study (Rev3).** We agree that ablation study without adversarial training is very important, as we claim the
permutation invariance to be an important property for good graph generation models. In fact, the first baseline GVAE in
our experiments is exactly GVGAN minus the adversarial training module, which is trained *w.r.t.* Eq.3 and referred to as
modified GVAE in the paragraph following Eq.3. We use this modified GVAE because the original GVAE (Variational
graph auto-encoders [Kipf et al. 2016]) is unable to generate graphs with flexible sizes and different semantic conditions
for meaningful comparisons with GVGAN. This is briefly mentioned in the **Baseline** section in experiments, and we
will definitely make it clearer in the final version.

[Meta-Review · NeurIPS 2019]

The work presents an interesting advance combining VAEs and GANs for graph generation as the reviewers all agree. The rebuttal for Reviewer 2's comments argue "random initialization works just as well, although initialization via spectral embedding may slightly help with the convergence 18 in model training and the robustness of the results". Please add numbers substantiating this in the camera-ready.